# Technical Development of the CeCi Social Robot

**DOI:** 10.3390/s22197619

**Published:** 2022-10-08

**Authors:** Carlos Flores-Vázquez, Cecilio Angulo, David Vallejo-Ramírez, Daniel Icaza, Santiago Pulla Galindo

**Affiliations:** 1Electrical Engineering Career, Research Group in Visible Radiation and Prototyping GIRVyP, The Center For Research, Innovation, and Technology Transfer CIITT, Universidad Católica de Cuenca, Cuenca 010107, Ecuador; 2Intelligent Data Science and Artificial Intelligence Research Center, Universitat Politècnica de Catalunya UPC BarcelonaTECH, Pau Gargallo 14, 08034 Barcelona, Spain; 3Laboratory of Luminotechnics, The Center For Research, Innovation, and Technology Transfer CIITT, Universidad Católica de Cuenca, Cuenca 010107, Ecuador

**Keywords:** social robotics, robot design, human robot interaction, mobile robotic platform, SLAM

## Abstract

This research presents the technical considerations for implementing the CeCi (Computer Electronic Communication Interface) social robot. In this case, this robot responds to the need to achieve technological development in an emerging country with the aim of social impact and social interaction. There are two problems with the social robots currently on the market, which are the main focus of this research. First, their costs are not affordable for companies, universities, or individuals in emerging countries. The second is that their design is exclusively oriented to the functional part with a vision inherent to the engineers who create them without considering the vision, preferences, or requirements of the end users, especially for their social interaction. This last reason ends causing an aversion to the use of this type of robot. In response to the issues raised, a low-cost prototype is proposed, starting from a commercial platform for research development and using open source code. The robot design presented here is centered on the criteria and preferences of the end user, prioritizing *acceptability* for social interaction. This article details the selection process and hardware capabilities of the robot. Moreover, a programming section is provided to introduce the different software packages used and adapted for the social interaction, the main functions implemented, as well as the new and original part of the proposal. Finally, a list of applications currently developed with the robot and possible applications for future research are discussed.

## 1. Introduction

The development of technology is one of the challenges for developing countries that regularly base their economy on the sale of commodities. This research aims to demonstrate the capacity of a country such as Ecuador to build its own technology and that this can be used for the development of new research in universities as well as for technology transfer for new products in the private sector. All those mentioned above could generate new qualified jobs in the country and an effect of economic welfare [1,2,3].

The development of this robot is mainly based on the article by Flores et al. [4]. However, the proposals of [5,6,7,8,9] were reviewed to complement and improve this prototype.

Considering the current market regarding existing equipment such as the Savioke Relay robot Figure 1a with a height of 0.91 m and a weight of 45.36 kg; with the ability to navigate autonomously indoors employing a LIDAR, 3D sensors, and several sonar sensors [10,11], which also enabled a container with 0.75 cubic feet of storage to fulfill functions of transport and delivery of elements within a hotel. The price of the Savioke robot is EUR 25,971.00 [12].

Others along the same lines as Savioke could be Vecna QC Bot, and Aethon TUG see Figure 1b,c deployed for environments such as hospitals and hotels [22]. The three previously mentioned have been on the market for several years, but their designs show a functional, task-oriented approach, which is far from the proposed user centered design approach.

The proposed robot CeCi is characterized by its utility as it is a service robot, but social design considerations have been emphasized to interact with users as described in [4,23,24].

Care-O-bot 4: this robot is a complete proposal considering the social aspects and functionality. It has a height of 158 cm and its full weight is 140 kg with 29 degrees of freedom (see Figure 1d) so it can be deduced that its cost is above one of the design conditions raised in this proposal, an accessible price for emerging countries [25]. A final consideration is that its arms can only hold 5 kg, so its constituent hardware generates an expectation that is higher than its useful functional capabilities. The price of Care-O-bot is EUR 65,927.00 to EUR 231,743.00 Euros depending on the configuration [26].

The Temi robot, which is manufactured by Robotemi, a company from Shenzhen, China, Figure 1e is a robot with various applications such as food delivery [27]. An application that stood out during the pandemic, its usefulness in medical assistance could be seen, as a thermometer and a thermal camera were added to its original design, mainly to deal with COVID problems and in telepresence activities in geriatric homes. Its basic configuration consists of four wheels, LIDAR, cameras, proximity sensors and an IMU. Among the considerations of this robot is its design, which does not generate empathy with people, is intended to be functional rather than social [28,29]. The TEMI price is EUR 7541.00 including travel and shipping case [30]. Import taxes of 12 to 25% should be added to this value.

Thalon Robot (Figure 1f) is the Colombian version of Savioke. Twenty engineers were employed for its development and construction, and the price is EUR 242,500.00. It is programmed to fulfill the functions of a minibar in hotels. It has refrigerated storage spaces in its torso. Navigation and obstacle avoidance are based on a 3D LIDAR, ultrasound and infrared sensor [31].

The Dinerbot–T8 from Keenon Robotics is one of the most refined robots in terms of aesthetic and Smart delivery capabilities. Features include centimeter-accurate positioning for athe smooth traversing of ultra-narrow paths, Intelligent Obstacle Avoidance, binocular vision plan with a 204${}^{\circ }$ real-time dynamic obstacle detection for safer and more flexible movement. Its height is 1096 mm [32]. See Figure 1g.

Pepper Figure 1h and Nao Figure 1i are two robots from Soft Bank Robotics. Unlike the previously reviewed robots with a task and practicality orientation, these prioritize the human-robot interaction approach. As such, their functionality or practicality is not a priority, and their ability to move objects and weight are lower than the previous robots. NAO’s price is EUR 19,478.00 [21] and Pepper Robot’s price is EUR 31,964.00 [33] both including a travel and shipping case. Pepper and Nao have allowed the development of interesting application cases in social robotics such as [34,35] that we wish to implement with CeCi in the near future.

This brief review of the main social-service robots allows to show the context described from the introduction concerning high prices and functionalities similar to the one of this research achieved by the CeCi social robot.

Even these robot prices increase with the addition of shipping costs, which is precisely why CeCi has straight and flat shapes to pack the disassembled robot in the smallest possible space and reduce transportation costs.

The patent number EC ECSDI22004944S process took place alongside the development of this paper [36]. The National Service of Intellectual Rights (SENADI) registers the patent in Ecuador. This organization defines four patent categories: Patent of Invention, Utility Model, Industrial Design, Layout-Designs of integrated circuits. CeCi is patented as an Industrial Design, specifically “the particular appearance of a product resulting from any meeting of lines or combination of colors, or from any two-dimensional or three-dimensional external shape, line, contour, configuration, texture, or material, without changing the destination or purpose of this product” [37]. The Figure 2 shows the nearest industrial robot designs registered in Ecuador by LG Electronics. However, the originality of the invention in the form of CeCi allowed its registration under the name of Social Robot. The registration of an industrial design will have a duration of 10 years.

### Social Focus

For the development of this research, the concept of acceptability was considered as a priority. “*Acceptability* as an evaluation before use or implementation of robots as opposed to acceptance, which is the evaluation after the implementation” [40]. The importance of human-robot interaction in social robotics lies in the matter of understanding ideas and preconceptions of the user, which is a vital element in the early stages of robot development [41].

Simple movements or gestures can give us relevant information about human activity [42] and this can be extrapolated to give intentionality of the activities that a robot might perform. In the case of CeCi, the eyes allow to achieve empathy for an adequate human-robot interaction and their verbal exchanges match the criteria of the articles [43,44].

The excellent article “What Makes a Social Robot Good at Interacting with Humans?” by Eva Blessing Onyeulo and Vaibhav Gandhi [45] they raise these two crucial questions: “Do social robots need to look like living creatures that already exist in the world for humans to interact well with them?”; “Do social robots need to have animated faces for humans to interact well with them?”. His answer to these questions is “that for humans to be able to interact well with a robot, the robot does not necessarily need to look like a human. However, the robot would benefit from some sort of “face”, a focal area that humans would have a direct conversation, especially if the facial features can move around to better express emotion in a life-like way”. This research ratifies CeCi’s proposed body shape that was achieved based on user feedback through focus groups.

In this proposal, the selection and operation of hardware and software is presented in Section 2, in Section 3, the results achieved with the CeCi robot, and finally, in Section 4, the objectives achieved and future work that could be developed with the robot.

## 2. Materials and Methods

This section will present the different constituent parts of the robot. In the the first subsection, a comparison of the sensors selected and implemented, an explanation of their operation and a comparison with other options will be made. The second subsection explains the implemented software and the capabilities that it allows the robot to develop.

### 2.1. Hardware Considerations

This section presents all the hardware components that passed the four filters imposed for this research: The first filter is that the hardware element (sensor, actuator, processor, case among others) meets the technical requirements explained in the tables of the finalist details in the selection. The second filter, which is the most restrictive, is that the price of this component must be the minimum possible while maintaining the standards of filter 1. The third filter is the availability of this part in the local market or that the supplier allows easy shipment to South America-Ecuador. Finally, but probably most importantly, that it is in line with the end user’s preferences. If the hardware component did not meet these three requirements it was discarded even though they stood out in any of the three criteria above the others. Therefore, in this section, only the components that met the four filter criteria are presented, explaining their final selection.

The mobile robot platform selected was iClebo Kobuki Figure 3, manufactured in Korea by Yujin Robot Co, Ltd. [46], because after a comparison with the iCreate 2 base of the manufacturer iRobot, it was found that Kobuki has more significant resources available in hardware and software in Robot Operating System (ROS) [47], so it has a greater possibility of action and versatility when testing for research and development purposes Table 1.

To process all the information received by the different peripherals, an Intel NUC i5 with 8 GB of RAM was installed in the first instance. Still, due to the amount of information processed simultaneously, especially when performing simulations in the GAZEBO application and visualizations in RVIZ, the machine would not respond and would freeze. Similarly, when using the camera functions in conjunction with the mobile robot base, the camera image was lost or turned off, due to the above, the equipment was upgraded to an INTEL NUC i7 with 16 GB of RAM. By making this change, a significant improvement was obtained. However, there is still no total fluency when running several functions of the robots simultaneously, especially simulations. For operation only with the physical robot the computer is adequate. If it is possible to increase the ram to 64 GB and add an external graphics card would be recommendable improvement.

In the camera section, a physical comparison was made between two options: Orbbec Astra camera and Intel RealSense D435i Figure 4, both of which fit very well with the robot. Considering mainly the range of vision, price, support and compatibility with the operating system used, the Orbbec was chosen. The improvements presented by the D435i in terms of image quality, weight, dimensions and even the inclusion of the IMU (Inertial Measurement Unit) [49], which is a helpful element with the capacity to measure and report the specifications of the orientation, speed and gravitational forces that the equipment can suffer, were insufficient in the cost-benefit analysis in Table 2. One problem to consider with the Intel camera is that it still has compatibility issues with the Kinect version of ROS, but not with the Melodic and Noetic versions.

When analyzing what type of laser scanner to use, three elements were taken into consideration. The first one was the Hokuyo UST-20LX which is an excellent piece of equipment with the best features and performance, but with a high cost for this application, it was discarded. So an option in line with a low-cost robot is the one offered by Slamtec. It was decided to install the RPLIDAR A3, its range, scanning frequency, and weight are sufficient for this robot and a considerable improvement over the previous version, the RPLIDAR A2, which was also considered as shown in Table 3.

For the screen, there are two GeChic portable touchscreen monitors with similar characteristics, the only difference being their size, 13.3 and 11.3 inches. The 13.3″ monitor was installed as it is considered to be better visualized due to its large size, but when carrying out tests it was noted that it presents complications due to its volume and weight, as it must be on top of the robot to be able to interact with people and this generates instability in the structure, which can be seen when moving along paths that are not wholly smooth. For future versions, the 11.3-inch monitor will be considered.

When testing in natural environments, the energy consumed by all constituent elements of CeCi, exceeded the capacity provided by the two batteries included in the Kobuki base. With this configuration, the robot’s autonomy in constant activity is 15 min and 25 min at rest. To solve this problem, an external 50,000 mAh Krisdonia battery Power Bank was included, which added autonomy of 4 h of continuous activity and 5 h and 30 min when idle. The external battery provides the power supply for the NUC computer, touch screen, lidar sensor, and camera. Therefore, the sensors and motors integrated in the mobile base are powered by the internal kobuki batteries, giving the mobile base an autonomy of 5 h of continuous activity on a flat surface and autonomy of 7 h at rest. Table 4 compares the batteries used, and Table 5 shows the consumption of the main peripherals of the robot. Figure 5 illustrates the hardware parts mentioned with their respective locations on the robot.

### 2.2. Software Considerations

Since the idea of making CeCi was generated, the open source platform Ubuntu 16.04 LTS was used, the version with long-term support [4]. During the course of the research several tests were carried out. One of them was to upgrade to 18.04 and even to version 20.04 but when installing and testing, it was found that it does not have compatibility with several packages necessary for the robot to fulfill the projected functions, so the development was continued in Ubuntu 16.04 LTS. Currently, CeCi software is supported on Ubuntu 20.04 LTS except for the navigation system, which is still in the process of being adapted. However, commercially available robots such as Temi, among others, are still using Ubuntu 16.04 for reliability.

It is crucial to consider the right environment for working in ROS, which is not an operating system in the most common sense of programming and process management. Instead, it provides a structured communication layer at a higher level than the host operating systems. So there are several structured versions and each is compatible with a different version of Ubuntu, in this case, the version compatible with the 16.04 LTS operating system is ROS Kinetic [51].

For a clear understanding of the software that enables the operation of the social robot, some remarks about ROS are necessary [52,53]: This framework provides services such as hardware drivers, device control, implementation of resources commonly used in robotics, message passing between processes and packet maintenance. It is based on a graph architecture where the processing takes place in nodes that can receive, send and multiplex messages from sensors, control, states, planning, and actuators, among others. These messages are called topics or services.

ROS is organized in terms of its software in (a) the operating system part, ROS, and (b) the ROS package that the user programmers contribute. Packages can be grouped into sets called stacks. Within the packages are the nodes to be executed. Figure 6 illustrates the above.

The Robotic Operating System (ROS) is free software that is based on an architecture where processing takes place in nodes [54] that can receive, send and multiplex messages from: sensors, control, states, schedules, and actuators, among others. The ease of integrating various functions [55] and utilities through its topical system (internal communication) [56] exemplified in Figure 6, makes it very appropriate for the development of algorithms and the progress of research in robotics-related subjects [57]. Although it is not an operating system, ROS provides the standard services of an operating system, such as: hardware abstraction, low-level device control, implementation of commonly used functionality, inter-process message passing, and package maintenance.

The CeCi software system is based on the structure provided by ROS in Packages, Nodes, Topics and Services. The following is an explanation by functional parts respecting the ROS architecture. The complete diagram of the Robot software was extracted from ROS using its graphical tool rqt_graph (node, topic and service diagram) and is included in Figure A1.

The main packages used for CeCi are Kobuki and Turtlebot 2 from the Yujim Robot manufacturer. There was no major problem because the ROS version used has support.

The Kobuki package allows to control and read the sensors of the mobile robotic base of the same name in detail, the utilities used are:Automatic return to its charging base;Publish bumpers and cliff sensors events as a pointcloud so navistack can use them;URDF and Gazebo model description of Kobuki;Keyboard teleoperation for Kobuki;A ROS node wrapper for the kobuki driver;Watches the bumper, cliff, and wheel drop sensor to allow safe operation;Set of tools to thoroughly test Kobuki hardware [58].

Figure 7 shows in a diagram the main functions contained in the package.

The closest robot to CeCi in terms of peripherals is the Turtlebot 2 [59], so this robot package was used as a basis. The Turtlebot package integrates all the functionalities of the Kobuki package plus those of the packages linked to the drivers and functionalities of the 3D camera (openni2_camera package) and the lidar sensor (rplidar package). In addition, special function (navigation) is used, which can be found in the turtlebot package developed by Gaitech. The Figure 8 illustrates the above.

This explains the robot capabilities for: (a) commanding and sensing its mobile base, (b) reading sensors and odometry including the lidar sensor and 3D camera Turtlebot Package, (c) navigation using the functions implemented within the Gaitech package.

The following explains the packages needed to perform navigation with Ceci. The first requirement is to have a 2D map of the environment in which you want the robots to navigate. A previously defined .yaml file can provide the map, or the robot can explore its environment and generate one.

To survey an environment in a 2D plane, two methods were implemented. The first one uses the Hector SLAM package [60,61] and the RPLIDAR A3 laser sensor. The second method implemented uses the SLAM Gmapping package and the Orbbec Astra 3D Camera as a sensor. Figure 9 illustrates the results obtained with each method, as it is evident that the first one gives better results.

Hector SLAM is an approach that adopts a 2D SLAM system based on the integration of laser scans (LIDAR) on a flat map and a 3D navigation system that integrates an inertial measurement unit (IMU) [62], to incorporate the 2D information from the SLAM subsystem as a probable source of information. It is based on optimizing the alignment of the beam ends with the map learned so far. The basic idea uses a Gauss–Newton approach inspired by the work in computer vision explained in [63]. Using this approach, there is no need for a data association search between the endpoints of the beam. As the scans are aligned with the existing map, the matching is implicitly performed with all previous scans in real-time triggered by the refresh rate of the laser scanning device. Both estimates are individually updated and flexibly matched to remain synchronized over time, giving precise results.

The SLAM Gmapping method focuses on using a Rao–Blackwellized Particle Filter (RBPF) [64] in which each particle contains an individual map of the environment, considering not only the robot’s motion, but also the robot’s current environment. This minimizes the uncertainty about the robot’s position in the step-by-step filter prediction and considerably reduces the number of required samples [65]. That is, it computes a highly accurate proposal distribution that is based on the observation probability of the most recent sensor information, odometry, and a matching process in the [66] sweep.

The package AMCL (Adaptive Monte Carlo Localization) is a probabilistic localization module which estimates the position and orientation of the CeCi robot in a given known map using a 2D laser scanner [67,68,69].

Each package fulfills a specific purpose and within them run nodes that contribute to the execution of that purpose. However, this does not limit the communication between nodes of different packages. Therefore all of them contribute to the complete task executed by the robot [70].

To control the navigation to specific points on a given map, a command file was formulated based on the map navigation file produced by Gaitech edu [71]. This script makes it possible to register coordinates as fixed points where the robot has to go. For this purpose, it uses its odometry and simultaneous localization [72], receiving orders or commands by different methods, either by keyboard, voice commands and mobile application.

Summarised from the navigation Figure 10, it can be understood from this short list of processes that CeCi fulfils the following points:1.The map of the previously surveyed site (map_server) is entered;2.The state of a robot is published in tf. When this is done the state of the robot is available to all components in the system that also use tf [72];3.The navigation command is executed by voice command, keyboard or mobile application;4.Probabilistic localization (AMCL) system for the robot to move, senses all its environment;5.Evaluates the best route to the target (Motion planning);6.Performs a diagnosis of the environment (Mapping), receives the command and the traced route to proceed to move the robot;7.While the robot follows the route the sensors keep working to avoid obstacles [73] and unevenness(SLAM) [66,74].

Orders received by the robot have three possibilities of input:Voice Commands: For the input of the information it is done through the package called Pocketsphinx, which is a continuous voice recognition system, it works independent of the speaker, based on discrete Hidden Markov Models (HMM) [75]. This system is based on the open source CMU Sphinx-4, developed by Carnegie Mellon University in Pittsburgh–Pennsylvania [76]. Through a Python script, sentences and commands are configured so that CeCi can perform movements and answers to the different preconfigured questions in the user’s language (Spanish). In order for CeCi to understand the commands dictated in Spanish, the training was taken from [77], since more than 450 h were invested in creating the language recognition files to form a complete dictionary, overcoming eventualities specific to Spanish.Keyboard: It is the minor complex of all, since only information is entered to a terminal through a keyboard connected by Bluetooth to the Intel Nuc computer inside the robot [78].Application: The ROS MOBILE application is a very intuitive and customizable open source user interfaces with the option to control and monitor the robotic system. In this case, a proprietary user interface was developed for Ceci with cursors and buttons for predefined motion functions. The connection is through a WiFi–Internet network between the robot and any mobile device (smartphones or tablets).

## 3. Results

In the previous sections the direction and methodology of this research were presented. This section shows a brief summary of the objectives achieved.

Regarding the physical components of the robot, Table 6 of characteristics summarizes them:

As mentioned in the introduction, one of the premises of this research is to achieve a prototype of a functional social robot, which can be made in an emerging economy country and therefore with a reduced price. Table 7 details the values of the components needed to make this robot. The sources consulted were amazon.com [79] and Ros Components [50]. For reference, we take the current conversion rate of USD 1 USD = EUR 0.93.

When 2D drawings of the environment in which the robot will navigate were made, it was verified that different results were obtained with the methods used. In the mapping comparison, considering the same conditions of speed in the movement of the robot, execution time and location of the robot in a controlled physical space, Figure 11a. As can be seen in Figure 11b, the plane obtained by the HECTOR SLAM method, has better definitions and resolution, generating an image with a width of 2048 pixels and height of 2048 pixels, unlike the plane obtained by GMAPPING that is observed in Figure 11c, which has the same range but less definition of the edges or limits, this method returns a file with an image of a width of 544 pixels and height of 480 pixels.

The main tested skills that can be compared to other social robots achieved with CeCi are:MappingNavigationSLAMVoice recognitionSpeechRemote OperationTelepresence

### Experimentation in Real Environments

Finally, some tests in natural environments are included to check the operation of the robot. In the Data Availability Statement section are links to four videos made about the robot.

Figure 12a shows the robot performing a service activity in a restaurant. Specifically, by lifting its hat, the customer can pick up two beers from inside.

Figure 12b shows an example of one of the uses of the robot where the dentist uses her touch screen to amplify the radiographies while performing patient care.

Gamification in learning applications is being explored in social robotics. Figure 12c shows the use of games on the CeCi touch screen to interact and generate gamified learning. The image corresponds to 3 and 4-year-old children.

Among the applications designed and realized with the robot is the delivery service of physical documentation in offices. Figure 12d shows a secretary inserting a folder into the robotic head for delivery to another office. It can also be seen that Ceci wears the uniform of the institution to allow identification with the staff. As part of future work is the process of patenting the clothing line for the CeCi social robot.

## 4. Conclusions

In this research, significant contributions were achieved for emerging countries such as Ecuador.

The first contribution lies in a quick introduction to the selection of the minimum hardware necessary for the implementation of a social robot with technical considerations. It is essential to clarify that there is existing hardware that was discarded from this analysis due to its high cost. In addition, a new perspective on the creation of robots centered on the user and his expectations using the Design Thinking methodology. As a result, the design patent was obtained.

The following contribution is a block diagram that summarises the conceptualization of the free software used to give it the required functionalities and which is not available in a complete form in state of the art, thus saving time for any future development of a social robot with similar requirements. These first two contributions could save development costs such as those mentioned in the development of the Colombian robot Thalos (EUR 242,500.00).

A differentiating factor of CeCi from the existing ones in the market is that it is not a robot specifically designed considering practicality and task, nor does it only consider social interaction is a mixture of the two according to the end user’s preferences.

In the construction and implementation of the robot, the hardware and software conceptualization presented were tested in a natural environment. Therefore, it is not only a theoretical or simulated proposal. As illustrated in the photographs in the results section and the supplementary material videos the robot has been tested in real environments (restaurants, offices, daycare centers, disinfection in hospitals with ozone, welcoming and attention to the public in events). One of the success stories of the robot was its participation in the local technology fair TEC BOX is aimed at entrepreneurs. As a result of this fair, the most important supermarket chain of the country wants the robot in its stores to welcome and inform customers. In addition to this company, there is another Deli chain that also likes it in its stores to help and promote customers. The contracts and tests with these companies will be carried out for one year, which is a more extended period than most tests with social robots.

As explained in this article, one of the objectives of this research was achieved, namely to achieve a low-cost social robot that is feasible to acquire and use in emerging countries such as Ecuador. It is essential to highlight that the research context in an emerging country is incipient, as demonstrated in the article (Scientific research in Ecuador: A bibliometric analysis) [80] and that it needs urgent strengthening with this type of research for its development. In detail, this prototype cost EUR 4191.51, and this cost could be decreased if its constitutive hardware is purchased in large quantities for its production. The relevance of making science and technology more affordable for countries with limited economic resources was one of the motivations for conducting this research.

The initial proposal was to generate a prototype that would allow research in the field of social robotics, hence its emphasis on cyclical design (prototype, evaluation, improvement) based on user criteria. The market segment addressed to this proposal were educational institutions, researchers, and developers. However, as explained, the result exceeded the expectations, and now even the commercial sector is interested as well.

Future work to be developed includes the publication of CeCi’s social interaction and its validation with people. Another major challenge is the long-term test with supermarket and deli companies. The system is entirely open in hardware and software, which allows exploring new applications with minor changes.

One of the solutions that CeCi uses to demonstrate its functions is its clothing, so a designer created a whole line of apparel for CeCi and her patent will be published in the next few weeks, probably one of the first social robots with a clothing line specifically created for its exclusive use.

## Figures and Tables

**Figure 1 sensors-22-07619-f001:**
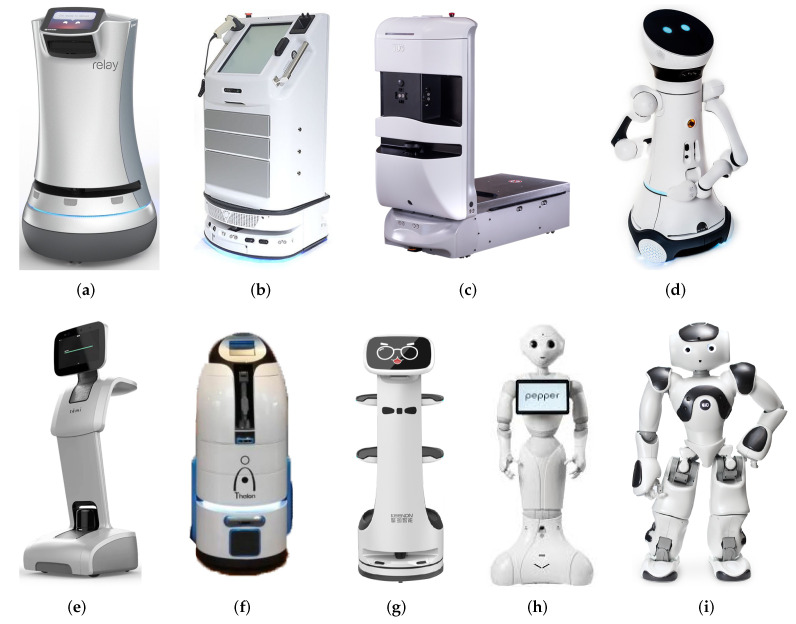
Robots: (**a**) Savioke [13] (**b**) Vecna QC Bot [14] (**c**) Aethon TUG [15] (**d**) Care-O-bot 4 [16] (**e**) Temi [17] (**f**) Thalon [18] (**g**) Dinerbot-T8 [19] (**h**) Pepper [20] (**i**) Nao [21].

**Figure 2 sensors-22-07619-f002:**
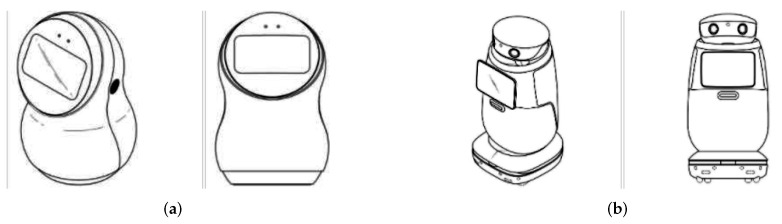
Domestic robot LG: (**a**) Patent 1 [38]. (**b**) Patent 2 [39].

**Figure 3 sensors-22-07619-f003:**
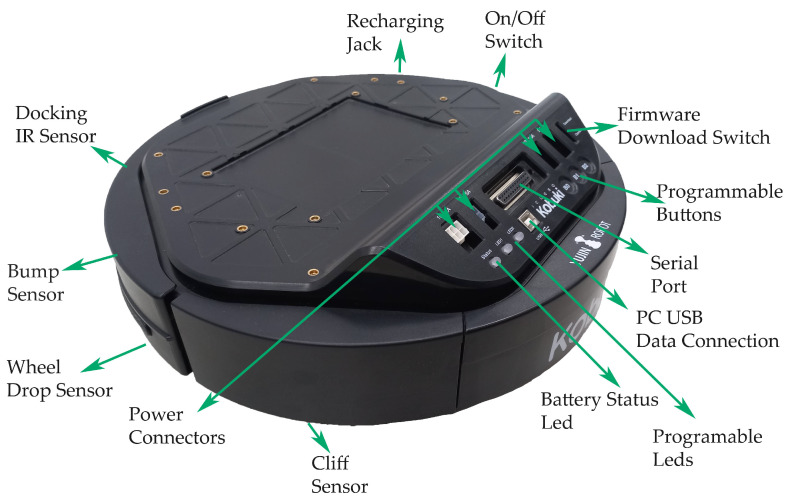
Kobuki Sensors.

**Figure 4 sensors-22-07619-f004:**
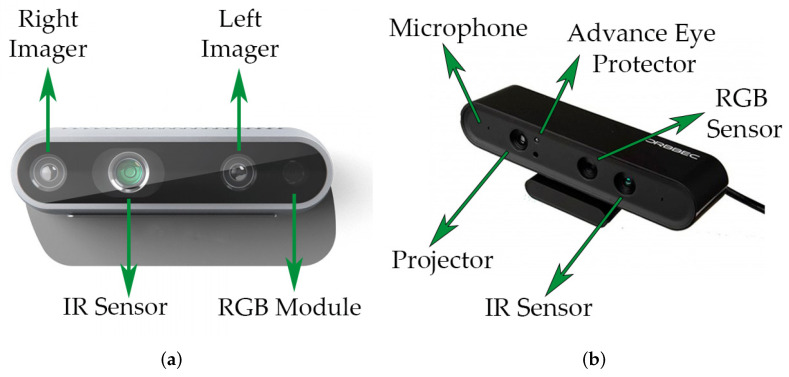
Cameras: (**a**) Intel RealSense D435i. (**b**) Orbbec Astra.

**Figure 5 sensors-22-07619-f005:**
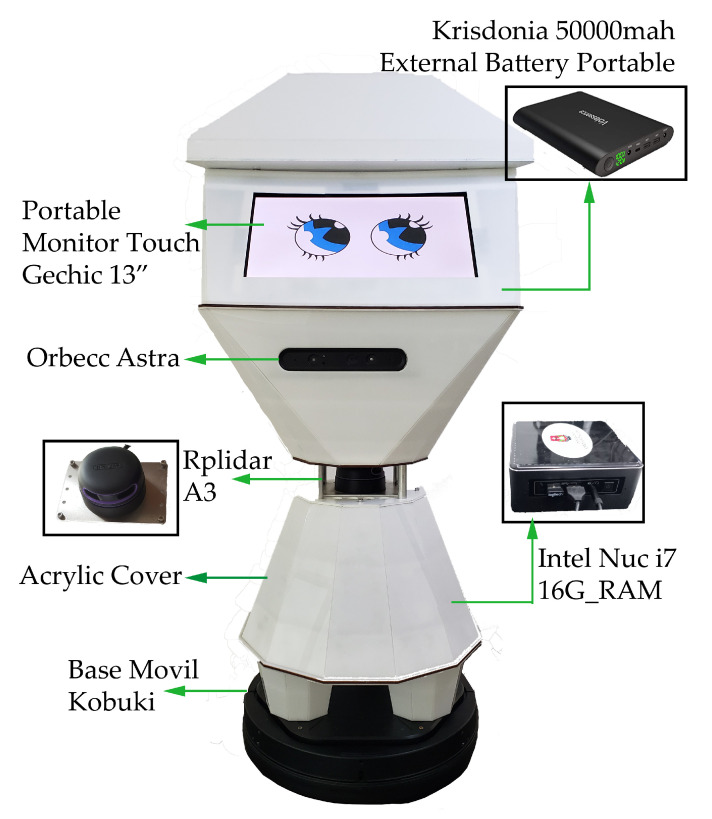
CeCi Parts.

**Figure 6 sensors-22-07619-f006:**
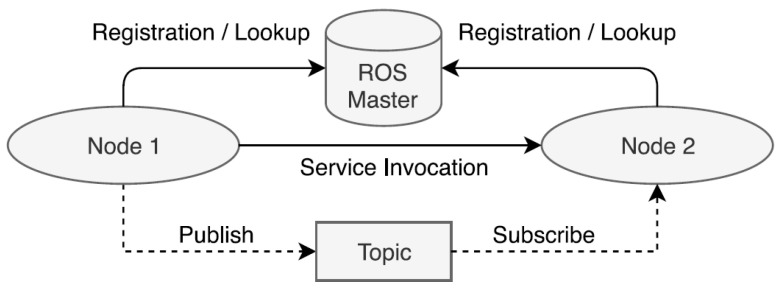
Graphical explanation of how ROS works in a brief overview [53].

**Figure 7 sensors-22-07619-f007:**
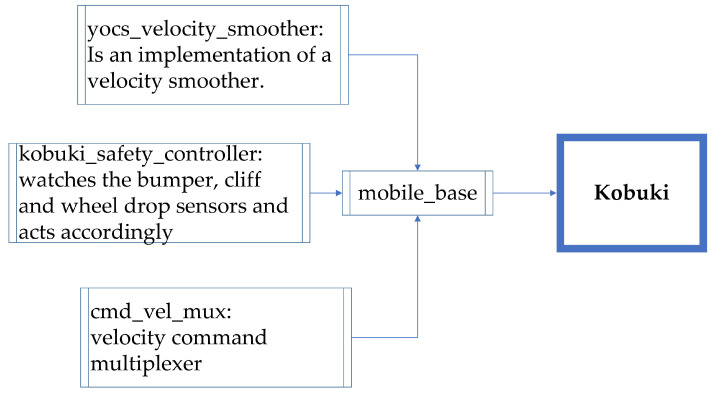
Kobuki block diagram.

**Figure 8 sensors-22-07619-f008:**
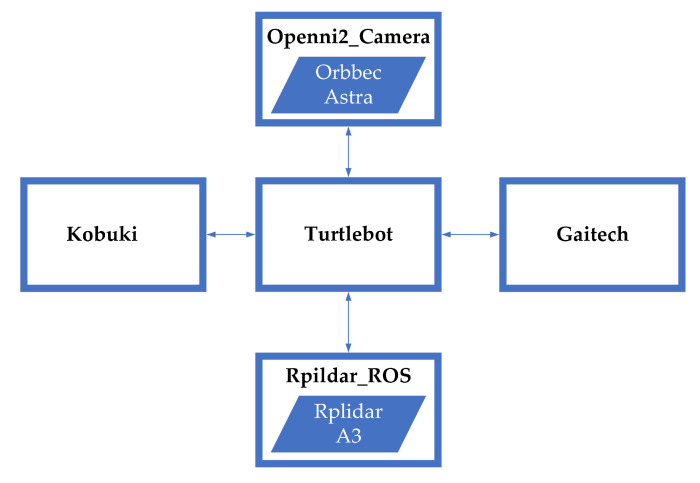
Turtlebot block diagram.

**Figure 9 sensors-22-07619-f009:**
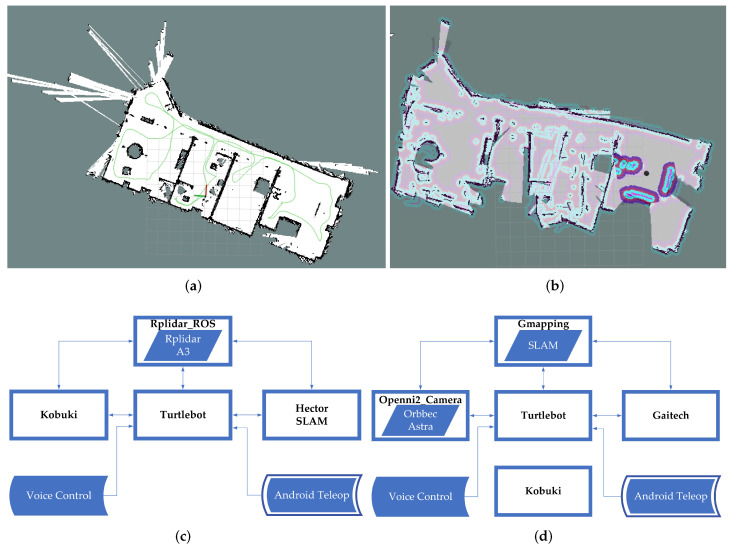
(**a**) 2D grid map with LIDAR. (**b**) 2D grid map with Camera. (**c**) Mapping Rplidar A3 block diagram. (**d**) Mapping Orbbec Astra block diagram.

**Figure 10 sensors-22-07619-f010:**
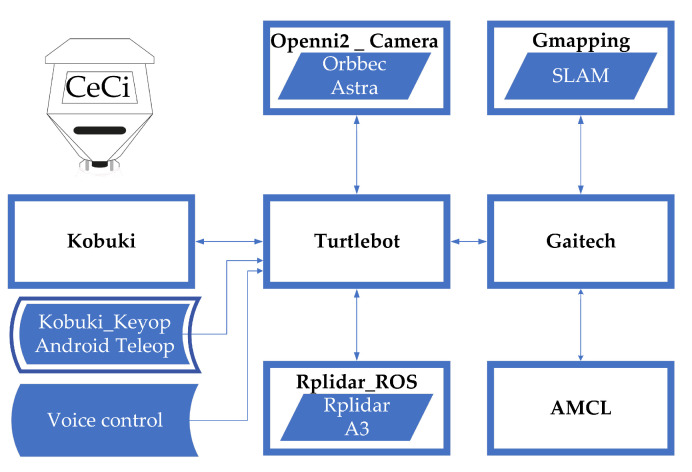
CeCi diagram.

**Figure 11 sensors-22-07619-f011:**
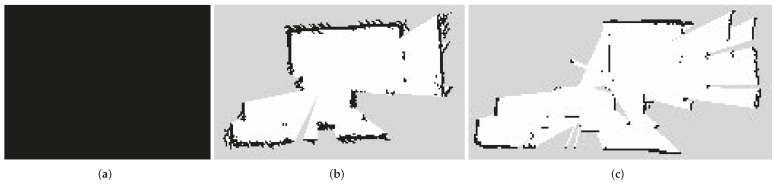
(**a**) Office. (**b**) Mapping of office with Hector SLAM. (**c**) Mapping of office with Gmapping.

**Figure 12 sensors-22-07619-f012:**
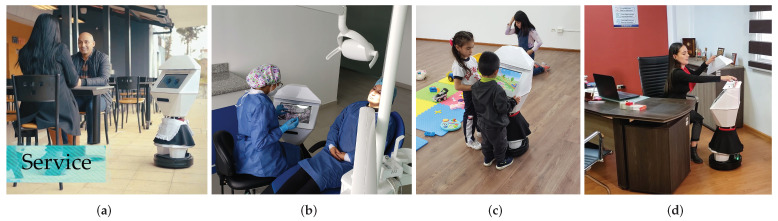
Testing the use of the robot in real environments: (**a**) Restaurant. (**b**) Dental office. (**c**) Kindergarten. (**d**) Office.

**Table 1 sensors-22-07619-t001:** Mobile Robot Base Comparison.

	Kobuki	iCreate 2 [48]
Odometry	Yes	Yes
Motor Overload Detection	Yes	No
Bumpers	Yes	Yes
Cliff Sensors	Yes	Yes
Payload	5 kg	5 kg
Battery	4400 mAh	4500 mAh
Price	649.00 €	200.00 €
Wheel drop sensor	Yes	No
Dimensions	Diameter: 351.5 mm/Height: 124.8 mm/Weight: 2.35 kg	Diameter: 340 mm/Height: 92 mm/Weight: 3.58 kg

**Table 2 sensors-22-07619-t002:** Camera comparison.

	Orbecc Astra	Intel RealSense D435i
Size	160 mm × 30 mm × 40 mm	90 mm × 25 mm × 25 mm
RGB Image Size	1280 × 960	1920 × 1080
Range	0.3 a 8 mts	0.2 a 3 mts
Field of View	60${}^{\circ }$ × 49.5${}^{\circ }$	86${}^{\circ }$ × 57${}^{\circ }$ (±3${}^{\circ }$)
Depth diagonal field of view over	73${}^{\circ }$	90${}^{\circ }$
Microphones	2	Not Available
Inertial Measurement Unit (IMU)	Not Available	Available
Reference cost [50]	EUR 149.00	EUR 349.00

**Table 3 sensors-22-07619-t003:** Laser Range Scanning Comparison.

	Hokuyo UST-20LX	RPLIDAR A3	RPLIDAR A2
Dimensions	70 mm × 35 mm × 50 mm	72.50 mm × 41 mm × 76 mm	72.50 mm × 41 mm × 76 mm
Weight	130 g	190 g	340 g
Angular Resolution	0.25${}^{\circ }$	0.225${}^{\circ }$ or 0.36${}^{\circ }$	0.45${}^{\circ }$–1.35${}^{\circ }$
Range	0.06–20 m	8–25 m	0.2–18 m
Precision	±40 mm	Not Specified	Not Specified
Sample Rate	40 Hz	15 Hz (adjustable between 5–20 Hz)	8 Hz
Reference cost [50]	2280.00 €	EUR 539.00	EUR 465.00

**Table 4 sensors-22-07619-t004:** Comparison of batteries used to power all components of the robot.

	Krisdonia 50,000 mah Power Pack External Battery	Integrated Kobuki Battery
Type	Lithium polymer	Lithium-Ion
Capacity	50,000 mAh	4400 mAh
Universal Compatibility	Yes	No
Fast Charging	Yes	No
Voltage/Amperage	5V/8.4V/9V/12V–3A; 16V/20V–4.7 A	3.3V/5V/12V/–1.5A; 12V–5A
Active Autonomy	4 h	15 min
Standby Autonomy	5 h 30 min	25 min

**Table 5 sensors-22-07619-t005:** Power consumption of the main components of the robot.

	Voltage (V)	Amperage (A)	Watts (W)
Orbbec Astra	5	0.38	1.85
RPLidar A3	5.5	0.6	3.6
Touchscreen 13.3″	5	1.6	8
Speakers, Mouse, Keyboard	5	0.6	3
NUCi7BNH	4.5	12	54
TOTAL			70.2

**Table 6 sensors-22-07619-t006:** Technical Specifications.

Datasheet	
Dimensions	990 mm (height)/360 mm (depth)/415 mm (width)
Weight	11 kg
Battery	Lithium-Ion: 4400 mAh (2 units) /Lithium polymer: 50,000 mAh
Camera	Orbbec Astra RGBD
Lidar	Slamtec RPLIDAR A3M1 360${}^{\circ }$ laser scanner
Display	GeChic 1306H Monitor Touch Display
Movil Base	Kobuki
Plataform	CeCi1.0
CPU	Intel^®^ Core™ i7-7567U Processor (4M Cache, up to 16.00 GHz)
Networking	Intel^®^ Wireless-AC 8265; Bluetooth 4.2; Intel® Ethernet Connection I219-V
Motion Speed	70 cm/s
Maximum rotational velocity	180 deg/s (>110 deg/s gyro performance will degrade)
Threshold Climbing	Climbs thresholds of 12 mm or lower
Odometry	52 ticks/enc rev, 2578.33 ticks/wheel rev, 11.7 ticks/mm
Bumpers	left, center, right
Payload	2.6 kg (hard floor); 1.2 kg (carpet)

**Table 7 sensors-22-07619-t007:** Price Table.

Item	Cost in Dollars (USD)	Cost in Euros (€)
Kobuki	571	531.03
Turtlebot2 Plate and Standoff Kit	200	186
RPLIDAR A3	600	558
Intel NUC 7 (NUC7i7BNH)	800	744
Krisdonia Battery 5000 mAh	130	120.9
GeChic 13.1’ Touch Screen	346	321.78
Kingston RAM 16GB DDR4	300	279
JBL GO2 Speaker	40	37.2
Microphone Lavalier	20	37.2
Body Construction	400	372
Others	100	93
Imports and taxes	1000	930
TOTAL	4507	4191.51

## Data Availability

Advertising TEC BOX video Robot Social CeCi: https://youtu.be/41rUGLJfSSk (accessed on 25 September 2022). Advertising press conference by Robot Social CeCi: https://youtu.be/qXYm40vscME (accessed on 25 September 2022). Interview for local TV media: https://youtu.be/Ao9Z9oT1ah0 (accessed on 25 September 2022). Mapping video Robot Social CeCi: https://youtu.be/cj5jWqOwlR0 (accessed on 25 September 2022). Navigation video Robot Social CeCi: https://youtu.be/ejuixFgRaTs (accessed on 25 September 2022).

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
