# Peer review of "Technical Development of the CeCi Social Robot"

_sensors, 2022, doi:10.3390/s22197619_

Round 1

Reviewer 1 Report

This paper mainly describes the technical specifications and functional realization of the hardware and software of a CeCi Social Robot, which has a certain practical value. It's easy to read for readers to understand, but there are a few problems.

In the introduction part of the article, the lack of analysis of the research background and problems to be solved, the significance of the study is unclear.

Keep the font consistent throughout the manuscript. (For example, the font in Figure 1-8).

Supplement with more experimental data to demonstrate the tested skills already achieved by the Ceci social robot.

Check the analysis of the paper to make it more logical and optimize the expression of the summary and conclusion.

Check the English language in the manuscript with special attention to the tense and word order of the manuscript.

Author Response

Dear reviewer, the answers and the revised article are in the attached file.

Reviewer 2 Report

In this paper, the authors consider the technical considerations for the implementation of the CeCi (Computer Electronic Communication Interface) social robot are presented. This robot responds to the need to achieve technological development in an emerging country with the aim of social impact, social interaction in this case. A low-cost prototype is proposed, starting from a commercial platform for research development and using open source code. This work details the selection process and hardware capabilities of the robot. The topic is quite interesting and this paper is well-written. I have the following comments for the further improvement of the paper.

1) The authors are suggested to remove the keyword "Covid- 19 robots" in the paper.

2) Some figures are unclear such as Fig. 6. Please provide a high resolution version of them.

3) Section 5 Patents can be presented in section 1.

4) The work mainly focuses on the technical implementation of robots. The reference part is suggested to be updated with some robot control methods to enhance related and previous work part, e.g., path-following control of autonomous underwater vehicles subject to velocity and input constraints via neurodynamic optimization; 

Tracking control of robot manipulators with unknown models: a Jacobian-matrix-adaption method;

5) The authors are suggested to present more future potential works in the conclusion part.

The paper is good writing and presents technical contributions, which could be accepted after a revision.

Author Response

(The authors gave the same response as above.)

Reviewer 3 Report

The manuscript describes the development of a social robot. A prototype built using low-cost hardware and open-source software is introduced. This reviewer has two main concerns regarding this manuscript:

1. From a scientific perspective, the manuscript has no contribution and adds no relevant information to the literature. The systems employed are already well described and validated in several papers, and their integration is used in a multitude of research and commercial robots. The rather scarce technical results are uninteresting.

2. From a practical perspective, the manuscript is not clear about its purpose nor about its design decisions. It reads as a report, with superficial description of parts and functionality. No requirements are presented and part selection is not detailed: it would seem that the authors choose from what would be available to them. Sentences as "In the camera section, a physical comparison was made between two options" (Line 71) reinforce this impression: why were these the options? Furthermore, the prototype's functionality is not demonstrated in any meaningful way.

Other comments are listed below.

- The Introduction must be deeply revised. The purpose of the proposed robot is not clear, and the literature and market review is too brief. Contributions are also not made clear.

- Why is the Kobuki compared against the iCreate? Without clear design guidelines, it seems that the authors compared any given two small robots and found out that one of them could be used in their system. Such a comparison and Table I are unnecessary. The same comment can be made about the other comparisons.

- Outdated software (Ubuntu 16 and ROS Kinetic) is used.

- Figure 4 is not well explained and, in my humble opinion, is unnecessary.

- Overall, the use of screen captures from ROS' rqt_graph should be avoided to favor more clear (functional) diagrams.

- Results section should be improved. It is merely a parts list with costs and a figure of well documented ROS-related results. There is also no experimental protocol.

- Conclusions are rather shallow, listing contributions that are not, in my humble option, actual contributions.

Author Response

(The authors gave the same response as above.)

Round 2

Reviewer 3 Report

Authors sufficiently addressed my main concerns with the manuscript and have properly answered all of my previous comments.